# Regulation of MAPK Signaling Pathways by the Large HERC Ubiquitin Ligases

**DOI:** 10.3390/ijms24054906

**Published:** 2023-03-03

**Authors:** Joan Sala-Gaston, Laura Costa-Sastre, Leonardo Pedrazza, Arturo Martinez-Martinez, Francesc Ventura, Jose Luis Rosa

**Affiliations:** Departament de Ciències Fisiològiques, Universitat de Barcelona, IDIBELL, 08907 L’Hospitalet de Llobregat, Barcelona, Spain

**Keywords:** ubiquitin, Large HERC, neurodevelopmental disease, cancer, MAPK, RAF, ERK, p38, PROTACs

## Abstract

Protein ubiquitylation acts as a complex cell signaling mechanism since the formation of different mono- and polyubiquitin chains determines the substrate’s fate in the cell. E3 ligases define the specificity of this reaction by catalyzing the attachment of ubiquitin to the substrate protein. Thus, they represent an important regulatory component of this process. Large HERC ubiquitin ligases belong to the HECT E3 protein family and comprise HERC1 and HERC2 proteins. The physiological relevance of the Large HERCs is illustrated by their involvement in different pathologies, with a notable implication in cancer and neurological diseases. Understanding how cell signaling is altered in these different pathologies is important for uncovering novel therapeutic targets. To this end, this review summarizes the recent advances in how the Large HERCs regulate the MAPK signaling pathways. In addition, we emphasize the potential therapeutic strategies that could be followed to ameliorate the alterations in MAPK signaling caused by Large HERC deficiencies, focusing on the use of specific inhibitors and proteolysis-targeting chimeras.

## 1. Introduction

Ubiquitylation is a post-translational modification that consists of the attachment of ubiquitin to a substrate protein. Ubiquitin can be linked as a single moiety or in the form of polymeric chains of different topology. Over the years, it has been discovered that proteins can also be modified by ubiquitin-like molecules such as SUMO or NEDD8. In addition, further modifications of ubiquitin with acetylation and phosphorylation have also been identified. Similar to a code, different ubiquitin modifications lead to different outcomes in the cells [1]. Typically, ubiquitin attaches to a substrate through the formation of a covalent bond between the C-terminal glycine residue of ubiquitin and an internal lysine residue of the substrate. However, in the last decade non-lysine ubiquitylation through ester linkages on serine, threonine and cysteine residues have also been established [2,3]. This expands the ubiquitin code with new layers of ubiquitin modifications and shows the complexity and potential versatility of the ubiquitin code in cell signaling.

Ubiquitin E3 ligases are the enzymes that catalyze the transfer of ubiquitin to a protein substrate. Therefore, they determine the precise substrate specificity of ubiquitylation, and play essential roles in the cell signaling networks mediated by ubiquitin [4,5]. According to their structure and mechanism of ubiquitin transfer, they can be classified into three main kinds: Really Interesting New Gene (RING) E3s, Homologous to the E6AP Carboxyl Terminus (HECT) E3s and RING-in-Between-RING (RBR) E3s [6].

RING E3s are the most abundant class. Up to 600 different types have been described in humans. They are characterized by the presence of a RING or U-box domain. Although U-box-containing E3 ligases have sometimes been classified as an independent class of E3s, sequence-profile analysis has shown that the U-box is actually a derived version of the RING finger, their mechanism of action being very similar [6,7]. These domains bind the ubiquitin-charged E2 enzyme and catalyze the transfer of ubiquitin directly to the substrate. Therefore, they act as a scaffold positioning the E2 enzyme in relation to the substrate protein [8].

Conversely, the HECT E3s catalyze ubiquitin transfer to the substrate protein by a two-step reaction. In brief, they present a HECT domain in their C-terminal region. This domain holds a bilobed structure that enables the transmission of the ubiquitin molecule to the target protein. Specifically, the first lobe (the one closest to the amino terminus) binds the E2 enzyme from which the ubiquitin is transferred to a catalytic cysteine located in the second lobe, forming a thioester bond. Next, the conjugation of ubiquitin to the target protein is catalyzed. In humans, 28 different types of HECT E3s have been identified [9].

RBR E3s represent the third class of E3 ligases. They are characterized by a mixed mechanism of ubiquitin transfer. They bind the E2 enzyme to a RING1 domain and transfer the ubiquitin to a second domain called RING2, which contains a catalytic cysteine. Then, the ubiquitin is transferred to the substrate protein. Despite being a small family, RBR E3s regulate different cellular process in human cells [10].

Centering now on HECT ubiquitin ligases, sequence and structural comparison analysis has shown a complex division of the HECT family into 16 different subfamilies: NEDD4-like proteins, Small HERCs, Large HERCs and 13 different subfamilies formerly called “other HECTs” [11]. In this review, we will focus on the HECT family of E3s and more specifically on the Large HERC subfamily, which is structurally characterized by the presence of the HECT domain in the C-terminal region and more than one RCC1-Like domain. This family is composed of two members: HERC1 and HERC2 [12].

Ubiquitin modifications regulate different cellular processes through proteolytic mechanisms, including targeting the substrate for proteasomal and autophagic degradation, but also through non-proteolytic mechanisms, for instance regulating protein interactions, subcellular localization and enzymatic activities. It is therefore not surprising that ubiquitylation is implicated in the regulation of several intracellular signaling pathways [13]. Some of these are the mitogen-activated protein kinases (MAPKs) signaling pathways. MAPKs constitute intracellular signal transduction cascades that in response to various extracellular signals elicit an appropriate intracellular response [14]. Each MAPK cascade consists of three core protein kinases: MAPKKK (MAP3K), MAPKK (MAP2K) and MAPK. MAP3Ks are the most upstream kinases and they activate the MAP2Ks by phosphorylation. In turn, MAP2Ks phosphorylate the downstream MAPKs, thus forming a three-tiered phosphorylation cascade [15]. Several HECT ubiquitin ligases have been implicated in the regulation of these pathways through different mechanisms [16]. In this review, we will give some insight into how HECT ubiquitin ligases can regulate MAPKs and more specifically, we will focus on the Large HERCs, since knowledge about their involvement in MAPK signaling is fairly recent and has grown in the last years.

## 2. Large HERCs and Disease

The physiological relevance of the Large HERCs is illustrated by their involvement in different diseases as shown in Figure 1. Although expressed in different tissues, Large HERCs are notably expressed throughout different areas of the nervous system [17]. Therefore, it is not surprising that most of the diseases with which these ubiquitin ligases are related are of a neurological nature.

Germ-line mutations in the *HERC1* gene are mostly associated with an autosomal recessive neurodevelopmental disorder called MDFPMR syndrome. This acronym comes from the phenotypic description of this disease, characterized by Macrocephaly, Dysmorphic Facies and PsychoMotor Retardation (OMIM #617011) [19,20,21,22,23]. In addition, genetic analysis has identified *HERC1* as a risk factor in the autism spectrum disorder, Parkinson’s disease, schizophrenia and febrile seizures [24,25,26,27,28]. Moreover, a missense mutation in mice that causes loss of cerebellar Purkinje cells, tremor and unstable gait also provokes myelin abnormalities in the peripheral nervous system, which are histological hallmarks of neuropathic periphery diseases [29]. Recently, *HERC1* was identified as a differentially expressed gene in the major depressive disorder associated with COVID-19 [30]. Other non-neurological diseases related to HERC1 include human immunodeficiency virus (HIV) acquisition and acquired immunodeficiency syndrome (AIDS) [31], diabetes [32], cardiovascular disease [33] and osteopenia [34] (Figure 1, HERC1 section).

On the other hand, germ-line mutations in *HERC2* are also associated with an autosomal recessive neurodevelopmental disorder (OMIM #615516). It was first identified in an Old Amish community holding a loss-of-function mutation in the *HERC2* gene (c.1781C>T, p.Pro594Leu). Later, other loss-of-function mutations have also been described. All cases show global developmental delay with Angelman syndrome-like features such as intellectual disability, autism spectrum disorders and movement disorders [35,36,37,38,39,40,41,42,43,44]. For further details, its phenotypic characteristics have been recently reviewed in [62]. In addition, due to the association between HERC2 and LRRK2 proteins, it has been suggested that HERC2 might be involved in Parkinson’s disease pathogenesis [45]. Other brain-related disorders in which HERC2 has been associated are agenesis of the corpus callosum (ACC), brain arteriovenous malformation (BAVM), diabetic cerebral ischemia–reperfusion (I/R) injury and central precocious puberty [46,47,48,49]. HERC2 has also been linked with other types of pathology such as refractive astigmatism [50], asthma [56], hypertension [57,58], several skin conditions such as vitiligo and rosacea [59,60,61] and some inflammatory diseases. Among these inflammatory diseases there are some related to the digestive system and, recently, HERC2 has also been described to promote inflammation-driven cancer stemness and immune evasion in hepatocellular carcinoma [51,52,53,54,55] (Figure 1, HERC2 section).

The involvement of both Large HERC family members in cancer has been extensively studied and it was previously reviewed in [18] (Figure 1).

Putting efforts into basic research to better understand how cell signaling is altered in these different pathologies is important to uncover novel therapeutic targets. To this end, in this review we will summarize the recent advances in how the HECT E3s and, more specifically, Large HERCs, can regulate the MAPK signaling pathways, emphasizing the potential implications in disease.

## 3. The Role of HECT Ubiquitin Ligases in the Regulation of MAPK Signaling Pathways

MAPK signaling pathways are intracellular signal transduction cascades that in response to various extracellular signals elicit an appropriate intracellular response affecting different cellular processes such as cell growth, cell proliferation, differentiation, migration, stress responses, survival and apoptosis. MAPK cascades can be activated by several factors such as hormones, growth factors, inflammatory cytokines and different types of stress [14]. Each MAPK cascade consists of three core protein kinases: MAPKKK (MAP3K), MAPKK (MAP2K) and MAPK, that in the classical three-tiered cascade are activated sequentially. In brief, in response to different stimuli, MAP3Ks are activated through phosphorylation, often as a result of their interaction with a small GTP-binding protein of the Ras/Rho family. MAP3K activation leads to the phosphorylation and activation of MAP2K, which then stimulates MAPK activity through phosphorylation. In turn, activated MAPKs eventually lead to the phosphorylation of target regulatory proteins in order to elicit an appropriate cellular response [15]. Currently, seven different MAPK cascades have been identified in mammals and named according to their central MAPK component. These are the so-called “conventional MAP kinases”, which include ERK1/2, p38, JNK and ERK5; and those termed “atypical MAPK kinases”, consisting of ERK3/4, ERK7/8 and NLK [16,63,64].

In recent years, it has become clear that the MAPK catalytic modules consisting of kinases that mediate the activation of downstream effectors are exposed to several layers of regulation. These regulatory mechanisms are not restricted to protein phosphorylation; instead, they involve other post-translational modifications such as ubiquitylation. Ubiquitylation catalyzed by E3 ligases influences MAPK pathways in terms of the duration and type of the signaling cascade through its role in affecting the assembly of protein kinase complexes, subcellular localization and the degradation of MAPK components or their downstream substrates [65]. For this reason, it is not surprising that several HECT ubiquitin ligases have been linked to these signaling pathways in different studies. Below, we will give a brief overview of the role of HECT E3s in MAPK regulation and, in the next section, we will focus on the Large HERC subfamily, as knowledge about their involvement in these pathways is fairly new.

In Table 1, we summarize the current scientific knowledge of HECT E3 enzymes that regulate the conventional MAPK signaling pathways, indicating the related molecular mechanism that has been described.

Some of them regulate a single MAPK pathway. For instance, NEDD4, UBE3A and HUWE1 regulate the ERK signaling pathway. NEDD4 regulates ubiquitylation of the G-protein coupled receptor (GPCR) mGlu7. Going into detail, after agonist stimulation mGlu7 is ubiquitylated by NEDD4, which induces its endocytosis and eventually leads to its degradation by both the lysosomal and proteasomal pathways. This mechanism is required for the mGlu7-induced ERK1/2 activation [69]. Moving now to UBE3A, in mice holding Ube3a deficiency, activation of ERK1/2 induced by membrane depolarization by KCl treatment is impaired. This suggests the presence of a regulatory mechanism between this HECT E3 and the ERK1/2 pathway, where UBE3A would act as an activator of this signaling pathway [84]. Finally, regarding HUWE1, it has been shown to regulate the ubiquitylation and protein levels of Shoc2, a C-RAF signaling partner. In turn, both Shoc2 and HUWE1 are required to control ubiquitylation and protein levels of C-RAF. In agreement with this, HUWE1 silencing increases C-RAF protein levels, which triggers ERK1/2 phosphorylation [85].

Other HECT E3 ubiquitin ligases have been implicated in the p38 signaling pathway. This is the case of HERC2 and UBR5. In both cases, their deficiency is related to increased levels of p38 activity, suggesting that they have an inhibitory role in this pathway. While the mechanism behind p38 regulation by UBR5 is not well established yet, HERC2 regulates p38 activity through controlling C-RAF ubiquitylation and its subsequent proteasomal degradation [68,86]. Lastly, in terms of the JNK signaling pathway, it has been described that the HECT E3 SMURF2 promotes ubiquitylation of tumor necrosis factor receptor 2 (TNF-R2). The formation of this ubiquitin chain appears to be required for TNF-R2-induced JNK activation [83].

There are also some HECT E3s whose involvement in more than one MAPK signaling pathway is currently described. For example, HERC1 regulates both ERK1/2 and p38 signaling pathways by modulating C-RAF protein levels through ubiquitylation [34,66,67]. Another HECT E3 that regulates both ERK1/2 and p38 signaling pathways is NEDD4L. NEDD4L overexpression is reported to inhibit ERK1/2 activation. In addition, NEDD4L regulates K63-linked ubiquitylation of protease-activated receptor 1 (PAR1), which leads to its activation and promotes recruitment of TAB2, which in turn associates with TAB1 and induces p38 activation independent of MKK3 and MKK6. Thus, while acting as a repressor of the ERK1/2 pathway, it would act as an activator of the p38 pathway [70,71,72].

Other HECT E3s have been shown to regulate three distinct MAPK pathways, including the ERK1/2, p38 and JNK cascades. In this regard, the ubiquitin ligase ITCH has been extensively studied for its multiple roles in the regulation of MAPKs, acting at different levels and affecting them in different ways in terms of activation or inhibition depending on the target and the context. Specifically, ITCH is recruited by GRAMD4 to promote ubiquitylation and protein degradation of the MAP3K TAK1, thus controlling activation of the conventional MAPKs ERK1/2, p38 and JNK [73]. In addition, K-27 linked ubiquitylation of the MAP3K B-RAF by ITCH leads to sustained B-RAF activation and subsequent elevation of the MEK/ERK1/2 signaling pathway [74]. Regarding the p38 pathway, ITCH acts inhibiting it, from one hand by targeting TXNIP for ubiquitin-dependent proteasome degradation, and from the other hand by catalyzing the K48-linked ubiquitylation and degradation of TAB1, a known activator of p38 [75,76]. In relation to the JNK pathway, ITCH has been described to regulate MKK4 ubiquitylation and subsequent degradation, thus decreasing JNK activation [77]. The HECT E3 ligases WWP1 and SMURF1 have also been related with the regulation of the three conventional MAPK signaling pathways ERK1/2, p38 and JNK. In cardiomyocytes, it has been reported that WWP1 regulates ubiquitylation and subsequent degradation of KLF15, which acts as an inhibitor of MAPK signaling. In particular, it was demonstrated that WWP1 overexpression down-regulates KLF15 and subsequently enhances ERK1/2 and p38 activation under hypoxic conditions [78]. Moreover, another study identified that under LPS stimulation, WWP1 binds TRAF6 promoting its K48-linked polyubiquitylation and subsequent proteasomal degradation. TRAF6 eventually leads to MAPK activation, hence, by means of increased protein levels of TRAF6, WWP1 knockdown activates ERK1/2, p38 and JNK pathways [79]. Regarding SMURF1, it seems to act, generally, as a repressor of the three conventional MAPK pathways ERK1/2, p38 and JNK. From one hand by interacting and modulating the ubiquitylation-mediated proteasomal degradation of MyD88, which is a major adaptor upstream protein of the Toll-Like Receptor (TLR) pathway [80]. From the other hand by promoting ubiquitylation and subsequent proteasomal degradation of the MAP3K MEKK2, a known activator of ERK1/2, p38 and JNK [81,82]. 

As shown in the table (Table 1) and indicated in the previous paragraph, the same HECT E3 can regulate different MAPK pathways by targeting a common upstream factor. For example, ITCH regulates ERK1/2, p38 and JNK signaling pathways by controlling ubiquitylation and protein degradation of the upstream kinase TAK1 [73]. Thus, there is some evidence that some HECT E3s regulate crosstalks between different MAPK cascades. Another example is HERC1, which regulates both ERK1/2 and p38 pathways via regulation of C-RAF protein stability. Thus, C-RAF, which has classically been defined as a MAP3K of the ERK1/2 pathway, appears that it may also act as a crosstalk factor for the p38 pathway, at least in a context of regulation by HERC1 [67]. Interestingly, the other Large HERC member, HERC2, also regulates p38 signaling through C-RAF [68]. This raises the question of whether this crosstalk of C-RAF to the p38 pathway is specifically regulated by Large HERCs. 

The precise molecular mechanisms of how the Large HERCs regulate MAPK signaling pathways through C-RAF are discussed in the following section. 

## 4. The Large HERC Ubiquitin Ligases in MAPK Signaling

Small and Large HERC subfamilies of the HECT ubiquitin ligases were initially classified together and defined by the presence of an HECT domain and RCC1-like domains [12]. However, it was demonstrated that the RCC1-like domains from Small and Large HERCs had significant phylogenetic differences [87], and that these domains were acquired by each subfamily in two independent events. Thus, the homology between Large and Small HERCs is due to a convergent evolution phenomenon rather to a common phylogenetic ancestor and, consequently, they should be classified in different protein subfamilies [11]. In addition, while Small HERCs only have one single RCC1-like domain, Large HERCs contain more than one in their structure. The Large HERC subfamily is composed by two members: HERC1 and HERC2. Due to their huge size they are designated as “Large” HERCs. HERC1 has 4861 amino acids with a molecular weight of 532 kDa, while HERC2 has 4834 amino acids and 528 kDa of molecular weight [12]. Structurally, HERC1 possesses the HECT domain, two RCC1-like domains (RLD1 and RLD2), an SPla and ryanodine receptor (SPRY) motif, a Bcl-2 homology domain 3 (BH3) and a Trp-Asp rich (W-D) 40-amino acid repeat region (WD40). HERC2, in turn, presents the HECT domain, three RCC1-like domains (RLD1, RLD2 and RLD3), a cytochrome b5-like region (Cyt b5), a mind-bomb/HERC2 (M-H) domain, a conserved domain within Cul7, PARC and HERC2 (CPH), a ZZ-type zinc finger region and a domain homologous to subunit 10 of APC (DOC) [9,12] (Figure 2a).

Knowledge about the involvement of the Large HERCs in the regulation of MAPK signaling is fairly recent and has evolved in the past years. In 2018 the involvement of HERC1 in the regulation of these signaling pathways was identified for the first time. More recently, a role for HERC2 was also described. Interestingly, both HERC1 and HERC2 appear to regulate MAPKs through the same protein target, the serine and threonine kinase C-RAF [66,68]. C-RAF, also known as RAF1, belongs to a family of protein kinases that has three members: A-, B-, and C-RAF. All three share some structural characteristics containing three conserved regions (CR1, CR2 and CR3). CR1 contains a RAS-binding domain (RBD) and a cysteine-rich domain (CRD). Importantly, CR3 contains the kinase domain [88] (Figure 2b). Through its kinase domain, C-RAF, which acts as a MAP3K, initiates the RAF-MEK-ERK1/2 phosphorylation cascade. The main function of this signaling pathway is to regulate cell growth, proliferation, survival and differentiation. For this reason, it is not surprising that mutations in C-RAF have been linked to cancer [89,90].

Both members of the Large HERC family interact with the C-terminal region of C-RAF, with residues 301 to 648 reported to be the most relevant. This region contains the kinase domain. HERC1 interacts through its N-terminal region, and the residues described as the most relevant are those from position 1 to 412. This region contains a small first portion of the RLD1 domain [66]. In contrast, HERC2′s affinity for C-RAF appears to reside in its C-terminal region. The region comprising the residues 4252 to 4834 seems to be the most relevant. This region holds the HECT domain [68,91] (Figure 2a,b).

Going into detail on the molecular mechanisms, HERC1 interacts with the MAP3K C-RAF and controls its protein stability by regulating its polyubiquitylation and subsequent proteasome-dependent degradation. Thus, HERC1 modulates the activation of the RAF-MEK-ERK1/2 pathway and controls cell proliferation [66]. Accordingly, HERC1 knockdown induces the stabilization of C-RAF, increasing its protein levels. In consequence, an overactivation of the RAF-MEK-ERK1/2 signaling cascade occurs. Remarkably, in these conditions of HERC1 deficiency, the p38 signaling pathway is also overactivated. The underlying molecular mechanism is that by controlling C-RAF protein levels, HERC1 also regulates the expression of MKK3, the MAP2K of the p38 signaling pathway. This crosstalk between ERK and p38 pathways occurs because the induced overexpression of C-RAF after HERC1 knockdown positively regulates the mRNA levels of MKK3 [66,67]. The physiological relevance of this crosstalk is that its activation affects cellular migration (Figure 3a). Therefore, the overactivation of ERK along with p38 that is triggered upon HERC1 deficiency could presumably lead to tumorigenesis and malignancy due to upregulation of the processes of cell proliferation and cell migration, pointing to HERC1 as a probable tumor suppressor protein [18,67,92].

The first evidence of the involvement of the other Large HERC family member, HERC2, in the regulation of MAPK signaling came from proteomic analysis where HERC2 was reported to be associated with E6AP, NEURL4 and the atypical MAPK ERK3 [93,94]. More recently, the participation of HERC2 in regulation of the p38 signaling pathway has also been described, expanding our understanding of the Large HERCs and their involvement in MAPK signaling. Similarly to HERC1, HERC2 interacts with C-RAF and controls its protein levels by regulating its polyubiquitylation-dependent proteasome degradation. However, although HERC2 knockdown increases C-RAF protein levels, this is not signaled through the RAF-MEK-ERK1/2 signaling pathway. Instead, it triggers MKK3-p38 cascade overactivation. This C-RAF-MKK3-p38 module induces the stabilization of the master regulator of the antioxidant response NRF2, which in turn enhances transcription of antioxidant genes such as *SOD1*, *SOD2*, *GPX1* and its own transcription through the *NFE2L2* gene (Figure 3b). This confers HERC2-deficient cells an overactivated antioxidant system, which causes an imbalance in the cellular redox homeostasis [68]. This mechanism could probably be exploited by tumor cells harboring HERC2 mutations, which would make cancer cells more resistant to the oxidative stress to which they are exposed [18,68]. A recent study also stablishes a regulatory link between HERC2 and NRF2. In this study, the authors described that *HERC2* gene holds putative antioxidant response elements (AREs) in which NRF2 is coupled to promote its transcription [95]. This suggests the presence of a feedback loop regulation in which HERC2 controls NRF2 stability through the p38 pathway and subsequently, NRF2 promotes HERC2 expression through enhancing its transcription. However, how this feedback is regulated and in which contexts it occurs still requires additional investigations.

It is interesting to note that although both HERC1 and HERC2 regulate MAPK signaling through modulating C-RAF protein levels, they diverge in the pathways that are finally modulated. I.e., while HERC1 regulates both ERK and p38 cascades, HERC2 regulation of C-RAF affects specifically the p38 pathway. Although the specific molecular mechanism by which these differences occur has not yet been described, there are some considerations important take into account. In first place, HERC1 and HERC2 proteins do not interact, suggesting that each of them forms a different protein complex with C-RAF. Secondly, while HERC1 interacts with the kinase domain of C-RAF through its N-terminal region, HERC2 does it through its C-terminal region, mainly involving its HECT domain (Figure 2a). Hence, the differences in cell signaling could be explained, at least in part, by the different complexes formed between C-RAF and each Large HERC protein [67,68]. 

Whether the alterations in these MAPK signaling pathways are associated with clinical outcomes in the neurodevelopmental disorders caused by mutations in HERC1 or HERC2 still requires further research. Even so, dysfunctions of the MAPK signaling pathways have already been implicated in several neurodevelopmental disorders, for instance autism spectrum disorder [96,97], which is also manifested in the syndromes caused by Large HERC mutations [17]. In addition, MAPK pathways are also associated with neurodegenerative diseases such as Alzheimer’s disease, Parkinson’s disease and amyotrophic lateral sclerosis [98,99]. All things considered, a deeper understanding of how Large HERC-dependent MAPK signaling affects the clinical manifestations of these neurodevelopmental disorders could reveal novel therapeutic approaches for these rare diseases.

## 5. Future Perspectives and Therapeutic Implications

Given that both HERC1 and HERC2 deficiency causes alterations in MAPK signaling due to increased levels of C-RAF, the most evident therapeutic strategy would be to inhibit C-RAF in order to block the triggered downstream overactivation. Indeed, the RAF inhibitors LY3009120 and sorafenib have shown success in counteracting ERK and p38 overactivation after HERC1 deficiency [66,67]. In the case of HERC2 deficiency, both inhibitors efficiently abrogate the increase in p38 phosphorylation caused by HERC2 downregulation and, importantly, they also reverse the overactivated antioxidant phenotype [68]. It is worth noting that sorafenib was approved for use in renal, hepatic and thyroid cancer in the USA in 2005 and in the EU in 2006 by the Food and Drug Administration (FDA) and European Medicines Agency (EMA), respectively [100,101]. Therefore, its use would represent a very feasible therapeutic option. However, experience with the use of RAF inhibitors in cancer has shown that there are some resistance mechanisms that eventually render cells insensitive to treatment (the RAF inhibitor paradox). The mechanistic basis by which this might occur is that RAF proteins form dimers. This implies that when RAF dimers are exposed to the inhibitors, binding of the drug to one monomer induces a conformational change that may result in the transactivation of the other non-drug bound monomer of RAF, and by this way bypass the inhibitory action [102,103]. With selective C-RAF inhibitors it seems that this phenomenon could also occur [104]. However, genetic ablation of C-RAF resulted in a complete regression of a subset of pancreatic carcinoma, inducing only some tolerable toxicities in adult mice [105]. These results suggest that other therapeutic approaches, aimed at the removal of the protein rather than in its inhibition, may provide greater therapeutic benefit.

In the last few years, an innovative technology called proteolysis-targeting chimera (PROTAC) has gained interest in the field of drug discovery [106]. PROTACs are heterobifunctional small molecules comprised of a moiety that links a protein of interest (POI), a linker and another moiety capable of recruiting an E3 ubiquitin ligase. By approaching the POI to an E3, the PROTAC enables polyubiquitylation of the target and its subsequent degradation by the proteasome [107]. The main advantages of PROTACs compared to the traditional inhibitory drugs are their increased selectivity, their ability to target previously “undruggable” proteins due to the fact that they do not necessarily target catalytic pockets, and their catalysis of the elimination of the target from the cell, which is pharmacologically more effective than a mere target inhibition [108]. In addition, PROTACs may also avoid the phenomena of resistance and transactivation that occur with the use of some inhibitors.

Taking this into account, could we use PROTACs to target MAPK alterations due to Large HERC deficiencies? So far, this approach has not been described nor studied in depth, but it might be a promising line of research. Since C-RAF is a common ubiquitylation target of the Large HERCs, and its accumulation is responsible for altered MAPK signaling in the case of HERC1 or HERC2 deficiency, one possibility could be to target C-RAF. In this aspect, several PROTACs against B-RAF, the most commonly mutated RAF isoform in cancer, have been successfully developed and provided promising results [108,109,110]. Hopefully, PROTAC technology will continue improving in the coming years, also expanding the number of target proteins to which they are directed, which may provide novel therapeutic opportunities.

## 6. Concluding Remarks

Large HERCs are involved in several diseases, with a notable implication in neurological diseases and cancer.HERC1 regulates ERK and p38 signaling pathways through controlling C-RAF protein levels.HERC2 regulates C-RAF protein levels, affecting the p38 signaling pathway.Downregulation of HERC1 or HERC2 causes accumulation of C-RAF protein levels which alters MAPK signaling.The use of RAF inhibitors such as sorafenib, or the development of specific PROTACs against C-RAF, may represent a promising therapeutic option to counteract alterations in MAPK signaling caused by HERC1 or HERC2 deficiency.

## Figures and Tables

**Figure 1 ijms-24-04906-f001:**
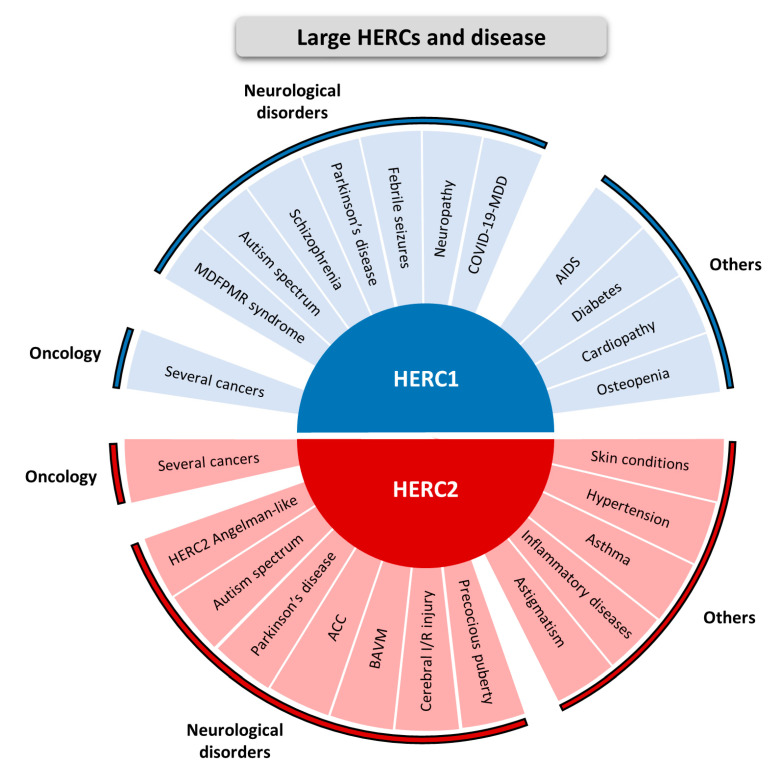
Large HERCs and disease. The different diseases that have been related to HERC1 (depicted in blue) and to HERC2 (depicted in red) are listed. HERC1: several cancers [18]; Macrocephaly, Dysmorphic Facies and PsychoMotor Retardation (MDFPMR) syndrome [19,20,21,22,23]; autism spectrum disorder [24,25]; Parkinson’s disease [26]; schizophrenia [27]; febrile seizures [28]; neuropathic periphery disease [29]; COVID-19 combined with major depression disorder (COVID-19-MDD) [30]; acquired immunodeficiency syndrome (AIDS) [31]; diabetes [32]; cardiovascular disease [33]; osteopenia [34]. HERC2: several cancers [18]; HERC2 Angelman-like syndrome [35,36,37,38,39,40,41,42,43,44]; autism spectrum disorder [17]; Parkinson’s disease [45]; agenesis of the corpus callosum (ACC) [46]; brain arteriovenous malformation (BAVM) [47]; diabetic cerebral ischemia–reperfusion (I/R) injury [48]; central precocious puberty [49]; refractive astigmatism [50]; inflammatory diseases [51,52,53,54,55]; asthma [56]; hypertension [57,58]; skin conditions [59,60,61].

**Figure 2 ijms-24-04906-f002:**
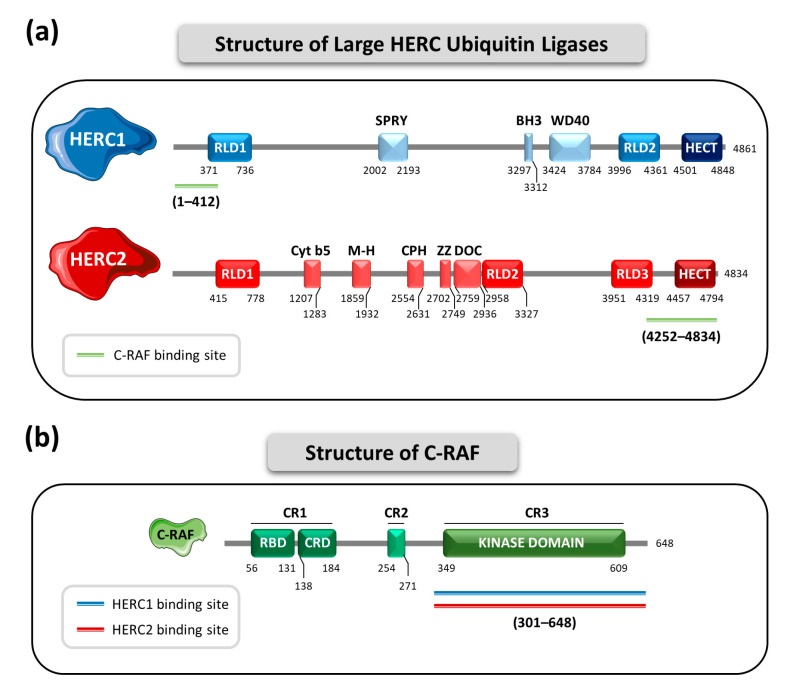
Structural characteristics of Large HERCs and C-RAF. (**a**) Domain architecture of HERC1 (depicted in blue) and HERC2 (depicted in red). Key domains and their boundaries are shown. C-RAF-interacting regions are indicated by a green line. (**b**) Domain architecture of C-RAF (depicted in green). Key domains and their boundaries are shown. The HERC1- and HERC2-interacting regions are indicated by a blue and red line, respectively.

**Figure 3 ijms-24-04906-f003:**
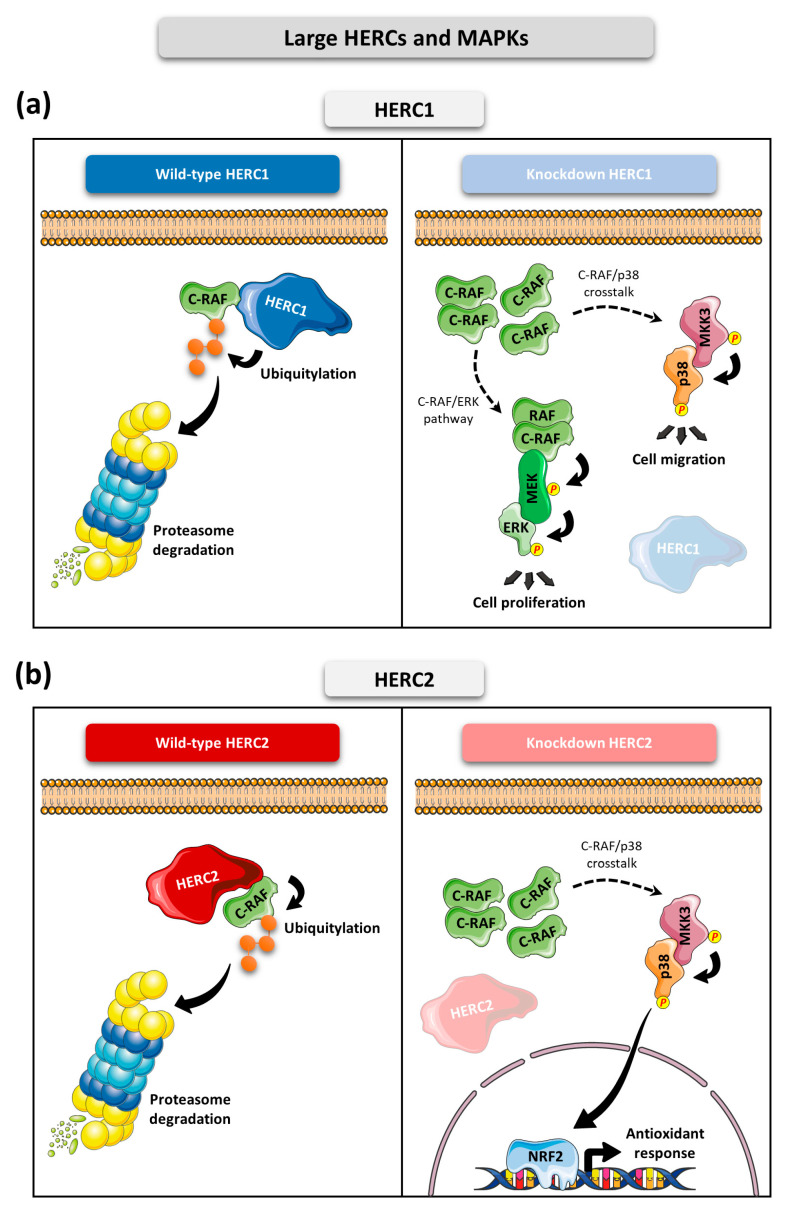
Large HERCs and MAPKs. (**a**) The N-terminal part of HERC1 interacts with the kinase domain of C-RAF, located in the C-terminus [66]. Through this interaction HERC1 regulates the protein levels of C-RAF via a ubiquitin-mediated proteasomal degradation mechanism. Knockdown of HERC1 causes accumulation of C-RAF protein. On one hand, this accumulation triggers overactivation of ERK signaling, thus enhancing cell proliferation. On the other hand, increased levels of C-RAF activate crosstalk between ERK and p38 pathways at the level of MKK3. Hence, p38 is phosphorylated and activated, which eventually promotes cell migration. (**b**) The C-terminal part of HERC2 interacts with C-RAF in its C-terminus kinase domain [68]. By this means, HERC2 controls protein levels of C-RAF by regulating its ubiquitylation and subsequent proteasomal degradation. Knockdown of HERC2 causes an increase in C-RAF protein levels. This C-RAF protein accumulation activates crosstalk between the C-RAF and MKK3/p38 signaling pathways which eventually induces the stabilization of NRF2 and leads to overactivation of the cellular antioxidant response.

**Table 1 ijms-24-04906-t001:** HECT E3 ubiquitin ligases regulating MAPK signaling cascades.

HECT E3s	MAPK Pathway	Mechanism	References
HERC1	ERK and p38	HERC1 regulates ubiquitylation-mediated degradation of C-RAF affecting ERK and p38 signaling	[34,66,67]
HERC2	p38	HERC2 regulates ubiquitylation-mediated degradation of C-RAF affecting p38 signaling	[68]
NEDD4	ERK1/2	Nedd4 regulates ubiquitylation and degradation of mGlu7, which mediates MAPK signaling	[69]
NEDD4L	ERK1/2	NEDD4L overexpression inhibits ERK1/2 phosphorylation	[70]
	p38	NEDD4L mediates K63-linked ubiquitylation of PAR1 inducing TAB1-mediated p38 activation	[71,72]
ITCH	ERK1/2, p38 and JNK	ITCH is recruited by GRAMD4 to target TAK1 ubiquitylation and degradation	[73]
	ERK1/2	ITCH ubiquitylates BRAF leading to its activation and subsequent elevation of MEK/ERK signaling	[74]
	p38	ITCH targets TXNIP for ubiquitin-proteasome degradation, decreasing p38 signaling	[75]
	p38	ITCH catalyzes the K48-linked ubiquitylation of TAB1, which modulates p38 signaling	[76]
	JNK	ITCH regulates MKK4 ubiquitylation and stability, affecting JNK signaling	[77]
WWP1	ERK1/2 and p38	WWP1 regulates ubiquitylation and subsequent degradation of KLF15, which acts inhibiting MAPK signaling	[78]
	ERK1/2, p38 and JNK	WWP1 modulates stability and LPS-induced TRAF6 ubiquitylation, affecting ERK, JNK, and p38 phosphorylation	[79]
SMURF1	ERK1/2, p38 and JNK	SMURF1 interacts and regulates ubiquitylation-mediated proteasomal degradation of MyD88, a MAPK signaling adapter	[80]
	ERK1/2, p38 and JNK	SMURF1 promotes MEKK2 ubiquitylation and degradation	[81,82]
SMURF2	JNK	SMURF2 induces TNF-R2 ubiquitylation and relocalization, which enhances JNK signaling	[83]
UBE3A	ERK1/2	UBE3A deficiency impairs ERK1/2 activation	[84]
HUWE1	ERK1/2	HUWE1 mediates ubiquitylation of C-RAF and regulates its stability, affecting ERK1/2 signaling	[85]
UBR5	p38	UBR5 silencing activates the p38 signaling pathway	[86]

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
