# Peer review of "Regulation of MAPK Signaling Pathways by the Large HERC Ubiquitin Ligases"

_ijms, 2023, doi:10.3390/ijms24054906_

Round 1

Reviewer 1 Report

Ubiquitylation of proteins by Ubiquitin E3 ligases is vital to normal cellular physiology. In this manuscript, the authors review historical and recent developments about one of the ubiquitin E3 ligases - HECT E3s (especially the large HERCs) and their role in regulation of MAPK signaling pathway. 

1. Line 25 - consists 'of'. 

2. Section 3, first paragraph (lines 127 to 143) seems like a deviation from the main topic of the review. A more concise version of this content could be included in the introduction where MAPK could be introduced to the readers after the HERCs. 

3. On a similar note, because the main topic here is large HERC and its regulation of MAPK pathways, rest of the content in section 3 seems less important than what comes in section 4. My recommendation is to discuss non large HERCs (i.e NEDD4, UBE3A, HUWE1 etc) at a later stage. 

Author Response

REVIEWER 1:

  1. Line 25 - consists 'of'. 

Thanks for the correction, we have corrected it.

  1. Section 3, first paragraph (lines 127 to 143) seems like a deviation from the main topic of the review. A more concise version of this content could be included in the introduction where MAPK could be introduced to the readers after the HERCs. 

Thanks for your suggestion. Now we have introduced a concise explanation of MAPK signaling pathways in the introduction (lines 73-88 in the version marked using the “Track Changes” function).

  1. On a similar note, because the main topic here is large HERC and its regulation of MAPK pathways, rest of the content in section 3 seems less important than what comes in section 4. My recommendation is to discuss non large HERCs (i.e NEDD4, UBE3A, HUWE1 etc) at a later stage.

We appreciate your comment. Our idea in this review was to give only a brief overview of the role of HECT ubiquitin ligases in MAPK regulation and then to focus on the “Large HERC” subfamily of the HECT ubiquitin ligases. This is why we explain Large HERC in more detail in section 4.

To clarify this and to make the content explained in section 3 more in line with the title of the section, we have made some changes:

  1. We have changed the title of section 3 to "The role of HECT ubiquitin ligases in the regulation of MAPK signaling pathways".
  2. We have added an explanation on lines 169-173, clarifying that we first give a brief overview of the HECT E3s, and then (in section 4) we focus specifically on the Large HERC subfamily of the HECT ubiquitin ligases.

Reviewer 2 Report

The paper is OK.

Author Response

REVIEWER 2:

The paper is OK.

We thank you for your comment and your positive feedback.

Reviewer 3 Report

The manuscript by Sala-Gaston and co-workers is interesting and summarizes the available information on the role of HERC ubiquitin ligases in the regulation of the MAPK signalling pathways. However, there are some concerns and comments.

According to the title and the abstract, the authors focus on the large HERCs, HERC1 and HERC2. However, not much information is available about these proteins yet, and there is a detailed summary concerning HECT E3 enzymes and their roles (lines 154-231, Table 1). Either the manuscript could be more general (focusing on HECT E3 enzymes) or more details about HERC1 and HERC2 would be needed (such as a scheme or figure about their structure).

The authors mention that targeting C-RAF may be beneficial. What are the functions of this protein? May its targeting alter other pathways resulting in side-effects?

What does NEDDylation mean?

Author Response

REVIEWER 3:

According to the title and the abstract, the authors focus on the large HERCs, HERC1 and HERC2. However, not much information is available about these proteins yet, and there is a detailed summary concerning HECT E3 enzymes and their roles (lines 154-231, Table 1). Either the manuscript could be more general (focusing on HECT E3 enzymes) or more details about HERC1 and HERC2 would be needed (such as a scheme or figure about their structure).

Thank you for your comment. Our idea in this review was to give only a brief overview of the role of HECT ubiquitin ligases in MAPK signaling. This is what we aimed with section 3 and Table 1. Next we wanted to focus on a specific subfamily of the HECT ubiquitin ligases, the Large HERCs, and explain them in more detail. For this reason we believe that the title should show this emphasis to the Large HERCs.

To clarify this, we have followed your suggestion and we have included more details about HERC1 and HERC2 structural characteristics, with an emphasis on the C-RAF binding domains (lines 255-294). In addition we have added a new figure to better illustrate this section (New figure 2).

The authors mention that targeting C-RAF may be beneficial. What are the functions of this protein? May its targeting alter other pathways resulting in side-effects?

Thank you for pointing out these concerns. In order to explain the functions of C-RAF in more detail, we have added a paragraph at the beginning of section 4 (lines 279-287).

Regarding the possibility of side effects from C-RAF inhibition, we have added some references and discussed them in section 5 (lines 385-396). In brief, the use of RAF inhibitors for cancer treatment has shown that, although being quite specific for the targeting of the ERK1/2 signaling pathway, therapeutic effects may be temporary due to the appearance of resistance (the RAF inhibitor paradox). For this reason, a different experimental approach aimed to induce protein removal instead of a mere inhibition might be more beneficial. For instance, genetic ablation of C-RAF resulted in a complete regression of a subset of pancreatic carcinoma, inducing only some tolerable toxicities in adult mice. Based on these precedents, we discuss the possibility of using PROTACs as a therapeutic approach.

What does NEDDylation mean?

NEDDylation is the process by which the ubiquitin-like protein NEDD8 is conjugated to its target proteins. To clarify this, we have added a small explanation in the introduction (line 27-30).

Round 2

Reviewer 1 Report

Suggested edits and changes have been made.